# Target Therapies for NASH/NAFLD: From the Molecular Aspect to the Pharmacological and Surgical Alternatives

**DOI:** 10.3390/jpm11060499

**Published:** 2021-06-02

**Authors:** Michele Finotti, Maurizio Romano, Pasquale Auricchio, Michele Scopelliti, Marco Brizzolari, Ugo Grossi, Marco Piccino, Stefano Benvenuti, Giovanni Morana, Umberto Cillo, Giacomo Zanus

**Affiliations:** 14th Surgery Unit, Regional Hospital Treviso, DISCOG, University of Padua, 31100 Padua, Italy; maurizio.romano@aulss2.veneto.it (M.R.); michele.scopelliti@aulss2.veneto.it (M.S.); marco.brizzolari@aulss2.veneto.it (M.B.); ugo.grossi@aulss2.veneto.it (U.G.); marco.piccino@aulss2.veneto.it (M.P.); giacomo.zanus@aulss2.veneto.it (G.Z.); 2Hepatobiliary Surgery and Liver Transplantation Unit, DISCOG, University of Padua, 35121 Padua, Italy; dottor.p.auricchio@gmail.com (P.A.); umberto.cillo@gmail.com (U.C.); 3Gastroenterology Unit (IV), Cà Foncello Regional Hospital, 31100 Treviso, Italy; stefano.benvenuti@aulss2.veneto.it; 4Division of Radiology, Treviso Regional Hospital, 31100 Treviso, Italy; giovanni.morana@aulss2.veneto.it

**Keywords:** non-alcoholic fatty liver disease, liver transplantation, bariatric surgery

## Abstract

Non-alcoholic fatty liver disease represents an increasing cause of chronic hepatic disease in recent years. This condition usually arises in patients with multiple comorbidities, the so-called metabolic syndrome. The therapeutic options are multiple, ranging from lifestyle modifications, pharmacological options, to liver transplantation in selected cases. The choice of the most beneficial one and their interactions can be challenging. It is mandatory to stratify the patients according to the severity of their disease to tailor the available treatments. In our contribution, we review the most recent pharmacological target therapies, the role of bariatric surgery, and the impact of liver transplantation on the NAFLD outcome.

## 1. Introduction

NAFLD (nonalcoholic fatty liver disease) is a metabolic disorder defined by the presence of hepatic steatosis (HS) and no other causes of hepatic fat accumulation (excessive alcohol intake, medication, or hereditary disorders). The presence of at least 5% of liver fat infiltration is necessary to define HS, either by imaging or histology. The HS alone characterized the nonalcoholic fatty liver (NAFL,) while nonalcoholic hepatic steatosis (NASH) and NASH-cirrhosis are defined as HS associated with hepatocyte injury (ballooning) with or without cirrhosis, respectively (see Figure 1) [1]. NAFLD is a condition that can be diagnosed through non-invasive tools, while a liver biopsy is necessary to differentiate NASH from NAFL. 

Globally, the prevalence of NAFLD is estimated to be around 25% to 30%, especially in the Middle East and South America. NAFLD is strictly correlated with relevant comorbidities, including obesity, dyslipidemia, hypertriglyceridemia, hypertension, type 2 diabetes, and metabolic syndrome. NASH is predicted to be the future leading indication of liver transplantation (LT) [2]. Obesity is a frequent condition associated with NAFLD, and its prevalence is increasing, present in around 28.6% of the U.S. population (90 million obese in a population of 315 million). In recent decades, the incidence is increasing as well, being higher in women (38.3%) than in men (34.3%). Worrying is the constant growth of obese children, which will likely represent the future population affected by NAFLD/NASH in more and more young people [3]. Using models based on current incidence, it has been estimated that by 2030 almost 40% of the world’s population will be overweight and 20% will be obese [4]. As a consequence, according to modern data and future projections, NAFLD is expected to increase to 33.5% among adults by 2030 (2), causing in the same time frame, a higher rate of hepatocellular carcinoma (+137%) and liver-related deaths (+178%) [5]. 

Recently, NAFLD is debated to be the correct nomenclature of the disease. Eslam et al. raised some concerns about the current definition [6]. First, NAFLD should not be a diagnosis of exclusion but inclusion. NAFLD is often associated with other liver diseases, such as viral hepatitis, autoimmune diseases, and alcohol, which influence differently the fatty liver progression and accumulation. 

Secondarily, the current definition leads to treating the disease as a single condition, not addressing the heterogeneity in risk profiles, the concurrent multiples comorbidities, and treatment responsiveness. So, to address the previous issues, the acronym MAFLD (Metabolic Associated Fatty Liver Disease) has been proposed to replace NAFLD/NASH [6]. However, as MAFLD definition is still not brought into ordinary use, in the paper we will use the most common NAFLD terminology. 

The aim of the paper is to review and categorize the broad spectrum of available therapies for NASH/NAFLD, based on its pathogenesis. In addition, we will evaluate the role of bariatric surgery, and the impact of liver transplantation on the NAFLD outcome. 

## 2. NAFLD Pathogenesis

The knowledge of the pharmacological targets comes from the understanding of the NAFLD pathogenesis. NAFLD genesis accounted for multiple factors. Liver inflammation and subsequent fibrosis are the key elements for NAFLD/NASH development. As for other diseases, the “multiple hit” hypotheses for the development of NAFLD have been proposed [7]. An unhealthy lifestyle, leading to excessive caloric intake, insulin resistance, gut dysbiosis, visceral fat mass, and increased hepatic de-novo lipogenesis and their interactions are all factors that have been related to NAFLD/NASH pathogenesis. Among them, insulin resistance seems to be an important element in the development of NASH. A genetic predisposition has been also proposed [8]. 

According to recent developments in molecular research, we can summarize NAFLD’s pathogenesis in three major fields, involving adipose tissue, nutrients, and intestine (See Figure 2). It is important to note that the knowledge of NAFLD’s pathogenesis is still under evaluation. 

It is well known that adipose tissue acts as an endocrine and immune organ. The production of adiponectin and other adipokines are involved in the development of NAFLD: adiponectin is an anti-inflammatory cytokine with anti-lipogenic effects which protect non-adipocyte tissue, such as the liver, from lipid accumulation [9]. Reduced levels of adiponectin are correlated with the development of NAFLD, obesity, and insulin resistance [10,11]. Other adipose tissue-derived signals, such as leptin and other cytokines (in particular IL-6 and TNF-α) are equally involved in the development of a systemic inflammatory status, enhancing steatosis and fibrosis. 

The nutrients could affect NAFLD development and progression in different ways. Other than the obvious caloric surplus, it is necessary to analyze how high levels of cholesterol and free fatty acids intake could enhance the pathologic process underneath NAFLD: -Palmitic acid and stearic acid can activate the intrinsic apoptotic pathway via C-jun-terminal Kinase and BIM, leading to mitochondrial permeabilization, the release of cytochrome c, and activation of caspase 3. Furthermore, palmitic acid and stearic acid lead to activation of the endoplasmic reticulum (ER) stress pathway, leading to apoptosis [12,13];-Oleic acid and palmitic acid activate BAX, which trans-locates to lysosomes, increases the permeability of lysosomes, and causes the release of cathepsin B, which further increases the permeability of mitochondria [14];-Ceramides are composed of sphingosine and fatty acid, and the availability of long-chain fatty acids is a rate-limiting step in the synthesis of ceramide in ER. In nutritional obesity with the associated elevation of palmitic acid and stearic acid, excess synthesis of ceramide is possible. Palmitic acid and stearic acid-induced de novo ceramide synthesis in a hematopoietic precursor cell line is associated with apoptosis [15,16];-Long chain polyunsaturated fatty acid (LCPUFA) oxidative stress leads to the depletion of n-3 LCUPFA (e.g., eicosapentaenoic acid, EPA, and docosahexaenoic acid, DHA) due to increased peroxidation or defective desaturation processes. Depletion of n-3 LCPUFA leads to the upregulation of lipogenic and glycogenic effects from SREBP-1c and down-regulation of fatty acid oxidation effects from peroxisome proliferator-activated receptor-α (PPAR-α), ultimately promoting hepatic steatosis [17,18,19].

The gut is implicated through the so-called gut–liver axis [7]. The liver receives more than 50% of its blood supply from splanchnic circulation, and hence, it is always exposed to gut-derived toxins. The ability of gut-derived factors such as lipopolysaccharide (LPS) to flow in the portal vein requires intestinal permeability, which in NAFLD is greater due to disrupted intercellular tight junctions in the intestine. Consequentially, plasma endotoxin levels are significantly high in patients with NAFLD [20], and a high-fat diet is associated with a 2- to 3-fold increase in plasma LPS levels. LPS could, therefore, activate an inflammatory cascade, involving stress and mitogen-activated protein kinases JNK, p38, interferon regulatory factors 3, and nuclear factor-jB. Those factors contribute towards insulin resistance, hepatic fat accumulation, obesity, and NAFLD/NASH development [21].

Obesity is strictly related to NAFLD pathogenesis, and weight loss through lifestyle modification is the first step in the disease treatment. The evaluation of what diet or exercises is most effective in the NAFLD treatment is not the aim of this paper. However, it is important to consider that less than 10% of patients affected by NAFLD can achieve a solid weight loss. For this reason, alternative therapies such as pharmacological and surgical options have been suggested. 

## 3. Current and Future Pharmacological Options

The principal endpoint in the randomized clinical trials (RCT), evaluating the pharmacological NAFLD treatment efficacy, is the resolution of the histological NAFLD hallmarks. The improvement or the regression of the hepatic steatosis, necrosis, inflammation, and/or fibrosis is the rationale of the currently available options. However, as previously reported, NAFLD is a multifactorial disease, and global treatment of the concomitant comorbidities, such as obesity, dyslipidemia, and/or diabetes is mandatory [22]. 

To date, there is no gold standard pharmacological prevention and/or treatment for NAFLD. The classification of the currently available options can be performed based on their beneficial effect on hepatic steatosis, inflammation, and fibrosis, with some agents having overlapping effects. Although this is not the aim of this review, it is important to understand that the correct way to evaluate the efficacy of these medications and their impact on the NAFLD improvement (NAS score, liver biopsy, steatosis measured with MRI tools) is an ongoing matter of debate [23].

### 3.1. Medications with Effect on Steatosis

#### 3.1.1. Statins

In the treatment of dyslipidemia, the role of statins is well recognized. In the beginning, the use of statins in patients with NASH/NAFLD raised some concerns due to their possible liver injury. Currently, the EASL/EASD/EASO guidelines considered statins safe, with no increased risk of liver damage. 

However, based on the current limited and conflicting evidence, the guidelines do not suggest routine prescription of statin in the treatment of NAFLD. To date, no completed RCT evaluated the role of statins in the NAFLD treatments. Most of the current data come from post hoc analyses of large RCT trials. 

In the GREACE study, in a sub-population of patients with features compatible with NAFLD disease, the use of statins resulted in a significant improvement in liver ultrasonography and liver function tests, associated with a reduction of a cardiovascular event [24].

Similar results were obtained in the post-hoc analysis of the IDEAL and ATTEMPT trials, which evaluated the impact of statins on populations with or without cardiovascular comorbidities [25,26]. The normalization in liver ultrasound and liver enzymes was reported and supported by other studies [27,28]. 

In studies with biopsy-proven effects, the use of statins showed a steatosis improvement and in some cases a NASH resolution, but no effect on the hepatic fibrosis [29,30,31,32]. 

Furthermore, studies suggested that statins are related to a reduction of hepatocellular carcinoma (HCC) risk and fibrosis progression [33,34,35]. A recent expert panel stated that the use of statins in NAFLD/NASH is safe and may contribute to slow down the evolution of NAFLD, reducing the morbidity and mortality related to a cardiovascular event and HCC [36]. 

#### 3.1.2. Orlistat

The obesity treatments are based on dietary and lifestyle modifications. Pharmacological options have been proposed, with data being limited or inconclusive. Orlistat is one of the most studied anti-obesity drugs. Weight loss is obtained from its ability to inhibit the gut and pancreatic lipases and reduce triglycerides’ absorption. 

In patients affected by NAFLD, orlistat effects are uncertain. Some studies reported no NAFLD improvement [37], others reported an improvement in the levels of alanine aminotransferase, aspartate aminotransferase, glucose, triglycerides, steatosis, and body mass index but not liver fibrosis score [38,39]. Other series reported an improvement in the hepatic fibrosis at biopsy evaluation, especially when associated with a significant weight loss. In this context, it is difficult to attribute the consequent improvement in hepatic steatosis and inflammation to the weight loss or orlistat treatment [40,41]. 

#### 3.1.3. Glucagon-Like Peptide 1 Receptor Agonists (GLP-1 RAs), Sodium-Glucose Co-Transporter-2 Inhibitor (SGLT2i), and DPP-4 Inhibitors

GLP-1 RAs acts on the pancreatic α-cells reducing the glucagon secretion and increasing the insulin secretion. An increase in fatty acid oxidation, a suppression/delay on the de novo liver lipogenesis, gastric emptying, and appetite are other important GLP-1 RA effects, especially in patients with type 2 diabetes T2DM and obesity [42,43].

In recent years, their application has been evaluated in NASH and NAFLD treatment, especially liraglutide. An improvement in steatosis and a slowing down of the fibrosis progression has been shown in the LEAN, LEAD, LEAN-J, and Lira-NAFLD trials. In the LEAN study, although in a small number of patients, a biopsy-proven NASH resolution was shown in 9/23 patients treated with liraglutide compared to 2/22 patients treated with placebo [44,45,46]. 

A recent phase 2b trial (NCT02970942) investigated the impact of another GLP-1 RA (semaglutide) in the NASH outcome. The primary endpoint, a resolution of NASH associated with no worsening in liver fibrosis, was met at the end of the study compared to a placebo [47]. 

SGLT-2i effects, especially empagliflozin and dapagliflozin, have been mainly investigated in the T2DM treatment. Their main effects are on blood glucose level control, thanks to their ability to reduce renal glucose reabsorption, increasing glycosuria, and promoting weight loss. In T2DM patients, SGLT-2i reduces cardiovascular risk and kidney injury. Most of the studies evaluated the SGLT-2i effect in patients with T2DM and NAFLD [48,49,50]. The E-LIFT trial showed a reduction of ALT and liver fat on liver MRI-PDFF [51].

A reduction of steatosis using the controlled attenuation parameter score was shown by Shimizu et al., which evaluated the effect of dapagliflozin in 57 patients affected by T2DM and NAFLD [52]. 

Using different methodologies to evaluate the liver fat (liver-to-spleen ratio on CT or MRI-PDFF), most of the recent studies confirm the ability of dapagliflozin to reduce liver fat accumulation, AST, and ALT levels [53,54,55].

An improvement of NASH, ALT/AST, intrahepatic triglyceride levels, and fibrosis using the fibrosis-4 (FIB-4) score has been shown with canagliflozin [56,57]. Ipragliflozin is another SGLT-2i associated with a reduction in liver stiffness, hepatic steatosis, and liver fibrosis on FIB-4 score [58,59]. 

To note, a post hoc analysis of the DURATION-8 study evaluated the effect of the association of GLP-1 RA with SGLT-2i (dapagliflozin and exenatide) in patients with DM2. The study showed a greater reduction in liver steatosis and fibrosis, using the fatty liver index and FIB-4, compared to dapagliflozin alone [60]. 

In conclusion, initial data showed that GLP-1RAs and SGLT2Is might reduce cardiovascular risk, NASH, and steatosis. However, most of the data derive from patients affected also by obesity and T2DM. The GLP-1 RAs and SGLT2Is effects on the NASH improvement can be related to the concomitant improvement on these comorbidities, especially weight loss [61,62,63]. Further randomized clinical trials with histological evaluation are needed to confirm these results.

DPP-4 inhibitors, used especially in the T2DM treatment, act on pancreas cells increasing insulin release and decreasing glucagon secretion. Their effects have been evaluated on patients with NAFLD, showing a non-consistent reduction in AST, ALT, and hepatocellular ballooning with no improvement on steatosis [64,65,66].

A recent meta-analysis evaluated the role of GLP-1 RAs, DPP-4, and SGLT on treating NAFLD in nondiabetic and T2DM patients. Twenty-one studies were considered, including three studies based on liver histology, showing that GLP-1 RAs and SGLT2 inhibitors decreased hepatic steatosis and steatohepatitis in NAFLD patients, while DPP-4 inhibitor therapy was not effective for patients with hepatic steatosis [67]. 

#### 3.1.4. Ursodeoxycholic Acid (UDCA)

Ursodeoxycholic acid has been associated with anti-apoptotic, anti-inflammatory, and hepato-protective effects, features which are particularly involved in NASH progression [68]. 

The UDCA effects, especially on the reduction of liver steatosis, have been proven in cell and animal models [69,70,71]. A systematic review evaluated the results from 12 randomized clinical trials on NASH patients, showing that UDCA uses, especially if associated with other antioxidants such as vitamin E, vitamin C, polyene phosphatidylcholine, glycyrrhizin, or tiopronin, was associated with improvements in liver function, steatosis, inflammation, and liver fibrosis [72]. 

However recent data showed the limited effect of UDCA. 

A recent randomized trial was conducted on 53 patients with biopsy-proven NASH. The patient was assigned to treatment with N-acetylcysteine (NAC) in combination with metformin (MTF) and/or UDCA for 48 weeks. A second biopsy was performed at the end of the study. In the intention-to-treat analyses, an improvement of steatosis, ballooning, and NAFLD activity score has been shown in the group treated with NAC + MTF, while no difference has been proven in the other treatments [73]. 

Another recent abstract showed the limited effect of UDCA compared to vitamin E plus C [74].

Thus, with the limited evidence to date, the standard use of UDCA in NASH patients is not supported, even if its use is associated with low adverse effects. 

#### 3.1.5. Mineralocorticoid Receptor Antagonists 

The effects of mineralocorticoid receptor antagonists (MRA), especially spironolactone, have been evaluated in the animal model and clinical field. In the mouse model with T2DM and NAFLD, MRA showed the ability to improve liver steatosis, insulin resistance, and to suppress inflammatory and lipogenic genes [75]. In the clinical scenario, a decrease in liver fat score and insulin resistance have been shown with low doses of spironolactone (25 mg/d) in association with vitamin E, especially after 1 year of treatment in patients affected by NAFLD [76]. 

However, the results are controversial. Recently, a study evaluated the effects of common medications on liver fibrosis among 1183 patients, 381 of them with biopsy-proven NAFLD and T2DM. Interestingly, patients who were consuming furosemide and spironolactone had a higher likelihood of having advanced fibrosis on liver biopsies. Probably, patients receiving these medications had advanced liver disease with ascites and fluid overload [77]. Further studies are needed, and an ongoing RCT is evaluating the effect of spironolactone in patients with NASH on liver stiffness, evaluated by magnetic resonance elastography (NCT03576755).

#### 3.1.6. Peroxisome Proliferator Activated Receptor (PPAR) Sparing

As we will describe later, thiazolidinediones (TZDs) are peroxisome proliferator-activated receptor (PPAR)-γ ligands. They are used as a second line for T2DM treatment and they were evaluated in the NASH treatment. However, they are associated with important side effects, some of them still under investigation. For this reason, an alternative to TZDs has been evaluated. 

Saroglitazar, a PPAR-α/γ dual agonist, and lanifibranor, a pan-PPAR agonist, are currently under investigation in ongoing RTCs (NCT03863574 and NCT03008070, respectively).

#### 3.1.7. Fibroblast Growth Factor (FGF-21) Analogs

Fibroblast growth factor (FGF-21), produced by the liver, has different effects on metabolic homeostasis: it enhances fatty acid oxidation, reduces hepatic lipogenesis, and enhances insulin activity improving peripheral glucose uptake. Some studies reported its anti-fibrotic effects in the liver [78,79]. In light of these effects, several FGF-21 analogs have been proposed. LY2405319 and PF-05231023 improved the lipid profile and reduced body weight [80,81]. 

Pegbelfermin, a PEGylated human recombinant FGF21 analog, has been evaluated in a phase 2a RCT in patients with NASH, showing a reduction in the hepatic fat accumulation [82]. However, to date, no RTCs evaluated its effect based on liver histology. Further studies are needed [83]. 

#### 3.1.8. Acetyl-CoA Carboxylase (ACC) Inhibitors (Firsocostat)

The enzyme acetyl-CoA carboxylase (ACC) acts in lipogenesis, converting acetyl-CoA to malonyl-CoA. Inhibition in the ACC activity has been speculated to reduce hepatic steatosis. In a pilot open-label prospective study considering 10 patients, firsocostat (an ACC inhibitor) showed a reduction in the hepatic de novo lipogenesis, steatosis (estimated by MRI-PDFF), and hepatic stiffness (measured by MRE) [84]. The ability of the firsocostat to reduce liver steatosis was confirmed by a subsequent study, but no change in LFTs and liver stiffness has been shown [85]. Recently, firsocostat has been evaluated alone or in association with the FXR agonist (cilofexor) and/or ASK1 inhibition (selonsertib). The primary endpoint, a ≥1-stage improvement in fibrosis without worsening of NASH, was not statistically significant; however, the data suggested that firsocostat, especially associated with cilofexor, resulted in greater improvements in histology and clinically relevant biomarkers versus either agent alone or placebo [86]. 

### 3.2. Medications with Effect on Fibrosis

#### ASK1 Inhibitors (Selonsertib)

ASK1 is a protein kinase that, after its activation by the tumor necrosis factors (TNF)-α, induces intracellular oxidative or endoplasmic reticulum stress, hepatocyte apoptosis, and fibrosis [87].

In the animal models, selonsertib, an oral ASK1 inhibitor, showed the ability to reduce hepatic fibrosis and steatosis [88]. In two randomized double-blind, placebo-controlled phase III trials, the anti-fibrotic effect of selonsertib was evaluated. Its effect was studied in patients with NASH and bridging fibrosis (STELLAR-3 trial) or compensated cirrhosis (STELLAR-4 trial). Liver biopsies were performed before and after the treatment, associated with noninvasive fibrosis tests. Both trials failed to show an improvement in the fibrosis without the worsening of NASH at week 48 [89,90].

### 3.3. Medications with an Effect on Steatosis and Inflammation

#### 3.3.1. Vitamin E

Oxidative stress is an important pathway in the development of NASH. Vitamin E, inhibiting the lipid peroxidation and the release of inflammatory cytokines, may improve liver inflammation and NASH. In the PIVENS (“Pioglitazone, Vitamin E, or Placebo for Nonalcoholic Steatohepatitis”) trial, patients with NASH treated with vitamin E (800 IU/day) showed an improvement in hepatic steatosis and hepatic inflammation, but not in fibrosis [91,92]. The histological improvement has been shown in non-diabetic patients, while in patients with NASH and T2DM (NCT01002547), Vitamin E alone did not reach the same results. Current guidelines recommend vitamin E 800 IU/day in non-diabetic patients with NASH [93]. 

However, the use of Vitamin E has to be limited in time, with relatively low doses, because studies have shown that prolonged high-dose vitamin E is associated with an increase in all-cause mortality in a dose-dependent way. Furthermore, Vitamin E has been linked to a higher risk of hemorrhagic stroke and prostate cancer, even if these results are inconsistent among the studies [23,94,95]. To conclude, the use of Vitamin E has to be tailored to patients considering his/her morbidity and risk factors [96]. 

#### 3.3.2. B Selective Thyroid Hormone Receptor (THR) Agonist (Resmetirom)

THR b, mainly expressed in the liver, is the prevalent b type of thyroid-stimulating hormone-releasing hormone (TRH) in the liver. Its activation, through a highly liver selective THRb agonist (resmetirom), is associated with liver steatosis and inflammation reduction [97]. 

A phase 3 study that evaluates the efficacy and safety of MGL-3196 (resmetirom) in patients with NASH and fibrosis (MAESTRO-NASH) is ongoing (NCT03900429).

### 3.4. Medications with an Effect on Inflammation and Fibrosis

#### C-C Chemokine Receptor (CCR2/5) Antagonist (Cenicriviroc)

In the animal model of NASH, chemokine receptors 2 (CCR2) and 5 (CCR5) are highly expressed. Their activation is associated with macrophage and monocyte stimulation, leading to hepatic stellate cell activation and subsequent fibrogenesis [23]. The inhibition of these receptors could lead to a reduction in liver inflammation and fibrosis [98,99]. 

Cenicriviroc (CVC), a dual CCR2/CCR5 inhibitor, has been recently evaluated in a phase 2b RCT. CVC has shown the ability to improve NASH in patients with severe disease (NAS ≥ 5). Furthermore, liver fibrosis is improved with CVC with low side effects [100]. A recent trial, the AURORA study, evaluated the efficacy and safety of CVC for the treatment of liver fibrosis in adults with NASH. The study was conducted in two parts: part 1 evaluated the fibrosis improvement (at least 1 stage) with no worsening of steatohepatitis after 12 months. Patients meeting the part 1 criteria would continue to part 2, which evaluated the long-term outcomes. However, the study was terminated early due to a lack of efficacy based on the results of part I (NCT03028740).

### 3.5. Medications with Overlapping Effects (Steatosis, Inflammation, and Fibrosis)

#### 3.5.1. TZD (Pioglitazone)

Thiazolidinediones are peroxisome proliferator-activated receptor (PPAR)-γ ligands and they are approved as a second-line treatment for T2DM. Their principal effect is on adiponectin regulation. Pioglitazone and rosiglitazone, the two most important TZD, have been evaluated in patients affected with NASH. The benefits are uncertain, with different outcomes based on different studies, and some potential important side effects have been shown [101]. 

The beneficial effects shown in some studies are an improvement in steatosis, lobular inflammation, and ballooning, with an uncertain role on the fibrosis improvement, especially in the short term [102,103,104].

The main controversies are about the possible side effects. Weight gain has been shown, and it is an important limitation for NASH treatment as most of the patients are obese. However, weight gain seems to be associated with a fat redistribution from visceral to subcutaneous adipose tissue that, on the other hand, could be beneficial [105]. Furthermore, pioglitazone has been correlated to a slight increase in myocardial infarction risk, bone loss density, and prostate, pancreatic, and bladder cancer risk, even if the correlation remains highly controversial, especially for prostate and pancreatic cancer [106,107,108]. Based on these data, pioglitazone should not be the treatment of choice in patients with bladder, prostate or pancreatic cancer and a high risk of fracture. The use of TZD can be used in NASH treatment after a careful evaluation and in selected patients until further data are available. 

A recent meta-analysis showed that pioglitazone associated with bariatric surgery is an effective treatment. In particular, pioglitazone and Roux-en-Y gastric bypass showed a reduction in non-alcoholic fatty liver disease activity score (NAS) [109]. 

#### 3.5.2. Selective PPAR-γ Modulator (SPPARMs)

Pemafibrate (K-877) is a recent selective PPAR-γ modulator (SPPARMs). In the mouse model of NASH, K-877 showed an improvement in the NAFLD activity score and ballooning degeneration [110]. Based on these encouraging results, an RCT with pemafibrate in NAFLD patients is ongoing (NCT03350165).

#### 3.5.3. Farnesoid X Receptor (FXR) Agonists (OCA)

Farnesoid X receptor (FXR) is expressed in the intestine and the liver, and they are receptors for bile acids. Together, their rule is to regulate lipid and glucose homeostasis [23]. 

Recently, farnesoid X receptor (FXR) agonist (cilofexor) has been evaluated alone or in association with ACC inhibitors (firsocostat) and/or ASK1 inhibition (selonsertib). The primary endpoint, a ≥1-stage improvement in fibrosis without worsening of NASH, was not statistically significant; however, the data suggested that cilofexor, especially associated with firsocostat, resulted in greater improvements in histology and clinically relevant biomarkers versus either agent alone or placebo [86]. 

Obeticholic acid (OCA) is another FXR agonist that demonstrated promising results, especially in liver fibrosis improvement, as shown in the phase III REGENERATE trial (NCT02548351). The main adverse reactions were pruritus, elevation in LDL cholesterol level, and an increase in gallstone formation and cholecystitis [23]. 

#### 3.5.4. PPAR Agonist (Elafibranor)

Elafibranor is PPAR-α/δ dual agonist, reducing the level of adiponectin [111,112]. In a recent randomized, double-blind placebo-controlled trial in patients with NASH, elafibranor was able to resolve NASH in patients with moderate or severe NASH. However, no or a limited effect has been shown in the fibrosis improvement. The treatment was well tolerated with limited side effects (i.e., mild and reversible increase in creatinine) but due to the limited results, the sponsors closed the trial (NCT02704403) [23].

### 3.6. Potential New Targets in the Treatment of NAFLD/NASH

Based on NAFLD/NASH pathogenesis knowledge, emerging new potential pharmacological targets are rising. Recently, in animal models, cholangiocytes associated with the secretin (SCT)/secretin receptor (SCTR) axis have been linked to NAFLD pathogenesis, playing an important role in biliary injury and hepatic fibrosis [113]. Another possible future target is the tumor necrosis factor-alpha-induced protein 3 (TNFAIP3). A recent study showed that, in animal models, TNFAIP3 and its modulation are associated with liver damage and inflammatory responses [114]. MiRNA has also been associated with lipid metabolism, hepatic inflammatory responses, and NAFLD progression [115]. Bone morphogenetic protein 8B (BMP8B), a member of the transforming growth factor-beta (TGFβ)-BMP superfamily, appears to be a major contributor to the liver inflammatory response. In vivo, the absence of BMP8B delayed NASH progression [116]. Apoptosis is another important step towards NASH progression. Recent studies showed that liver inflammation, injury, and steatosis might be improved by inhibitors against this pathway [117,118,119,120]. The immune system has also been linked to NASH progression, in particular, an auto-aggression of CD8 T cells within the liver [121]. All these potential new pharmacological targets are based on the NAFLD/NASH pathogenesis, which is still under evaluation.

## 4. Bariatric Surgery for NAFLD and NASH

To improve liver function and observe a consistent change in NAFLD, a sufficient weight loss in the case of obesity is mandatory. In particular, a 5% weight loss can improve liver steatosis, while a 7% and 10% weight loss is necessary to achieve liver inflammation and fibrosis improvement, respectively. 

However, since weight loss is not always easily feasible, bariatric surgery has proven to be the best approach [122].

The current indications for bariatric surgery are: -Body mass index (BMI) of 40 or higher (extreme obesity);-BMI from 35 to 39.9 with a serious weight-related health problem, such as type 2 diabetes, high blood pressure, or severe sleep apnea [93]; -In some cases, patients could be qualified for weight-loss surgery if BMI is between 30 and 34 with serious weight-related health problems [123];

Different studies showed that the typical liver features (steatosis, fibrosis, ballooning degeneration) in obese patients with NAFLD after bariatric surgery improved [124,125]. 

In 2019, Lee et al. described through a meta-analysis biopsy-confirmed study how bariatric surgery resolves steatosis in 66% of patients (95%; CI 56–75), inflammation in 50% (95%; CI 35–64), ballooning degeneration in 76% (95%; CI 64–86), and fibrosis in 40% (95%; CI 29–51) [126]. 

As described in a case-control study by Wirth et al., bariatric surgery could be performed not only for the actual indications [93], but also to reduce the liver injury and progression. The study showed that in the two years after bariatric surgery the overall risk of developing cirrhosis decreases [127].

However, the data are controversial. In Lee’s meta-analysis, 12% of patients developed a worsening of fibrosis after bariatric surgery, despite the weight loss. A possible explanation could be related to which bariatric procedure fits better than others for NAFLD patients.

The Roux-en-Y gastric bypass (RYGB) is the most common bariatric surgical procedure worldwide, introduced more than 60 years ago. It is also one of the effects on NAFLD that have been most studied [128]. Most of the studies indicate that RYGB is able to improve the severity of NAFLD [129]. 

However, a sleeve gastrectomy (SG), as described by Weiner, offers good results within two years after surgery in the NAFLD natural history, proven by an improvement in liver function test as a surrogate marker [125,130]. 

RYGB and SG may significantly improve alanine transaminase, aspartate transaminase, NAFLD activity score, and NAFLD fibrosis score, but direct comparisons among them failed to demonstrate the superiority of one procedure versus the other [131]. Further studies evaluating the effect of bariatric surgery in liver disease progression are needed, especially to understand which patients benefit most from this type of surgery and what kind of procedure fits best. Furthermore, the interactions and the best combination of the available pharmacological therapies and the surgical options are still a matter of debate. 

## 5. Liver Transplantation and NAFLD

Since Powell et al. described the progression of NAFLD towards cirrhosis [132], in most cases the unclear definition of “cryptogenic cirrhosis” turned out to be advanced NAFLD, as Caldwell et al. stated [133]. These patients are at a high risk of progressing into end stage liver disease, requiring LT. NAFLD has been retrospectively identified as the underlying cause in 30 to 75% of cryptogenic cirrhosis [134,135].

In the large variety of LT indications, NAFLD is mostly seen in older patients, probably due to a slow fibrosis progression, and a long, silent course. Patients could experience a liver failure as the first presentation (38 to 45% of cases) and once cirrhosis decompensates, hepatic deterioration develops in a short time, driving patients in the urge of LT [136].

Furthermore, a growing number of publications have linked insulin resistance, NAFLD, cryptogenic cirrhosis, and HCC [136,137]. Although cirrhosis per se is a preneoplastic condition, both obesity and T2DM are recognized risk factors for HCC, irrespective of the presence of cirrhosis [138].

### 5.1. NAFLD as an Indication for LT 

According to a rising prevalence of NAFLD, and the strong connection between HCC and end stage liver disease, LT is an important therapeutic chance for these patients. NAFLD is now the second most common etiology of chronic liver disease in the United States [139]. 

In the years to come, NAFLD will probably become the main indication for LT because of: (1) the worldwide increasing prevalence of NAFLD paralleling the increasing prevalence of metabolic syndrome, diabetes, and obesity; (2) the absence of a valid noninvasive diagnostic tool to allow the early diagnosis of the disease, leading to the under-recognition of NAFLD before the cirrhotic stage; (3) the absence of therapies that can effectively prevent disease progression; (4) the new direct-acting antiviral era and the possibility to cure HCV resulting in stabilization or a decreasing of the number of cases of HCV-related liver failure, allowing LT for other indications.

### 5.2. Timing of Bariatric Surgery and Effects of NAFLD on the Waiting List for LT

According to the 2013 OTPN/scientific registry of transplant patients (SRTR) annual data report [139], the proportion of obese (BMI ≥ 30 kg/m^2^) patients undergoing LT significantly increased over the past decade from 28% to 35.5%, with 3.6% of patients having a BMI > 40 kg/m^2^. 

Several aspects should be particularly considered for NAFLD patients listed for LT, especially in obese patients [140,141,142]. 

Morbid obesity (BMI ≥ 40 kg/m^2^) is often considered a relative contraindication for LT. To address this problem, different approaches have been proposed to combine bariatric surgery and LT. As described by Diwan and Rice [140], the strategy of using bariatric surgery in the peri-transplant period could facilitate the enrollment of the patient for LT. Bariatric surgery, especially SG (easier to perform and with lower post-operative complication compared to RYGB) can be performed before or concomitantly to the LT, reducing the BMI and visceral adipose tissue. Bariatric surgery may affect the survival, since the rapid improvement of metabolic parameters could balance the lipidic and glucose impairment caused by immunosuppressant drugs (e.g., corticosteroids, tacrolimus). 

Some considerations due to the peculiarity in the management of this population in the LT waitlist and peri/post-operative period should also be addressed: -Patient with a MELD score of less than 15 usually has a slow progression of NAFLD, resulting in a longer waiting list period (annual progression rate of 1.3 vs. 3.2 MELD points in NAFLD vs. HCV patients) [143];-NAFLD patients frequently dropped out from the waiting list because of associated comorbidities, older age, impaired renal function, and lower MELD [143]. Adjusting for the MELD score, the short and long-term survival (90 days and 1 year, respectively) on the waiting list was lower in NAFLD than in alcoholic liver disease [144,145]; -Obese patients with NAFLD have a significant increase in post-operative complications, related to sarcopenic obesity, requiring longer hospitalization and specific management [146];-Portal vein thrombosis is associated with more complicated surgical procedures, increased post-transplant mortality and morbidity, and, if extensive, may lead to patient drop out from the waiting list for LT [147]. A recent analysis of the UNOS/OPTN database found a higher prevalence of portal vein thrombosis in patients with NAFLD cirrhosis when compared with other etiologies [148] probably due to a procoagulant imbalance [149]. -Patients with NAFLD are more likely to develop the “small for size syndrome” and, therefore, are less likely to be eligible for living donors and split LT. 

A chronic, silent, obesity-related disease, such as NAFLD, affects mostly patients with a higher BMI, higher prevalence of T2DM, metabolic comorbidities, lower glomerular filtration rate, and most of all, old patients: this represents a consistent problem on LT waiting lists and the patient management before and after the LT [144,150,151,152].

## 6. Discussion and Conclusions

NAFLD is an increasing cause of morbidity and mortality, especially in the Western area. This disease is often associated with other important comorbidities, such as obesity, hypertension, and DTM2, which makes its treatment particularly complex. To date, there is no gold standard pharmacological prevention and/or treatment for NAFLD, due to different reasons. First, the mechanisms underlying NAFLD are still only partially understood and pharmacological therapies are still not standardized. Second, many suggested drugs failed to improve the liver histological hallmarks of NAFLD. Third, a current problem is its early diagnosis. NAFLD is usually asymptomatic and, to date, we lack non-invasive tools that can accurately recognize the disease. As for other diseases, early detection is essential, in order to treat the condition before its complications and progression. Fourth, different therapeutic options are available, according to the stage of the disease. In particular, the treatment options can range from lifestyle modification, pharmacological options, and bariatric surgery ending in LT in selected cases. However, the general understanding about current treatments (pharmacological, surgical, or both) is that we lack evidence of significant benefit in terms of NAFLD improvement. It could be reasonable, from our perspective, to anticipate medical and surgical treatments usually started in later phases of the disease, such as bariatric surgery in the setting of clinical trials.

These treatment options are not standardized, and the selection of the best ones and their interactions can be challenging. All of the available treatments should be individually tailored to the patient’s needs and comorbidities, with a multidisciplinary approach that considers lifestyle modification, and the current pharmacological and surgical treatment options.

## Figures and Tables

**Figure 1 jpm-11-00499-f001:**
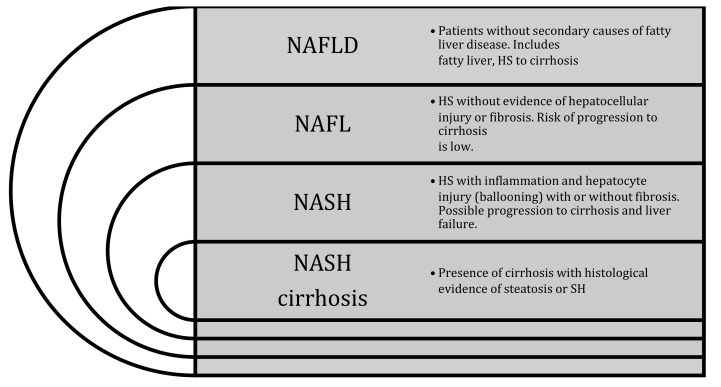
Nonalcoholic fatty liver disease (NAFLD), nonalcoholic fatty liver (NAFL), and nonalcoholic hepatic steatosis (NASH) definitions [1].

**Figure 2 jpm-11-00499-f002:**
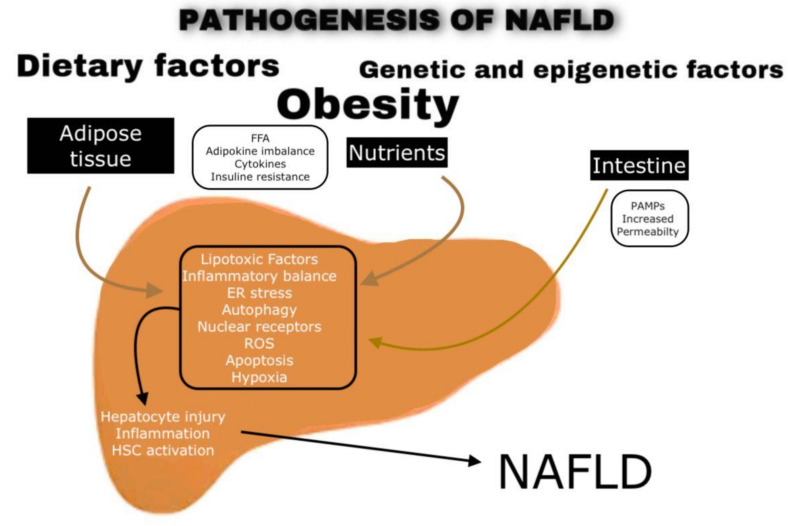
The role of the adipose tissue, nutrients, and intestine in the pathogenesis of NAFLD.

## Data Availability

Not applicable.

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
