# Peer review of "Target Therapies for NASH/NAFLD: From the Molecular Aspect to the Pharmacological and Surgical Alternatives"

_jpm, 2021, doi:10.3390/jpm11060499_

Round 1

Reviewer 1 Report

Review article submitted by Finotti and colleagues discusses target therapies for NASH/NAFLD: from the molecular aspect to the pharmacological and surgical alternatives. The authors define the major medications that are currently investigated to treat NAFLD/NASH.

Overall, this review covers most of the relevant literature and discusses the most recent pharmacological target therapies in a comprehensive and systemic fashion. However, the review will benefit from a paragraph resuming the potential new targets in NAFLD/NASH.

For instance, the necroptotic pathway has been recently investigated, and it has been shown that inhibitors against this pathway may reduce not only inflammation and cell injury but also steatosis (PMID: 31132314, PMID 31760070, PMID 32094147). Another example is bone morphogenetic protein 8B (BMP8B), a poorly characterized member of the BMP–TGFβ superfamily, which appears as a major contributor to NASH progression (PMID: 32694734)

Author Response

Thank you for the comment. As suggested, we added a section:

3.6 Potential new targets in the treatment of NAFLD/NASH

Based on NAFLD/NASH pathogenesis knowledge, emerging new potential pharmacological targets are rising. Recently, on animal models, cholangiocytes associated with the secretin (SCT)/secretin receptor (SCTR) axis have been linked to NAFLD pathogenesis, playing an important role in biliary injury and hepatic fibrosis[113]. Another possible future target is the tumor necrosis factor alpha-induced protein 3 (TNFAIP3). A recent study showed that, in animal models, TNFAIP3 and its modulation are associated with liver damage and inflammatory responses[114]. MiRNA has been also associated with lipid metabolism, hepatic inflammatory responses and NAFLD progression[115]. Bone morphogenetic protein 8B (BMP8B), a member of the transforming growth factor beta (TGFβ)-BMP superfamily, appears to be a major contributor to the liver inflammatory response. In vivo, the absence of BMP8B delayed NASH progression[116]. Apoptosis is another important step towards NASH progression. Recent studies showed that liver inflammation, injury and steatosis might be improved by inhibitors against this pathway[117-120]. The immune system has also been linked to NASH progression, in particular, an auto-aggression of CD8 T cells within the liver[121]. All these potential new pharmacological targets are based on the NAFLD/NASH pathogenesis, which is still under evaluation.

Reviewer 2 Report

Can you please elaborate the introduction section to shed some more light on the problem, its causes, the burden on society, and current challenges in treatments along with the significance of the review? 

Can you add a discussion section where you can perhaps discuss the advantages and drawbacks of the treatment methods and provide a personal outlook of the future of treatment methods? 

Author Response

Can you please elaborate the introduction section to shed some more light on the problem, its causes, the burden on society, and current challenges in treatments along with the significance of the review? 

Thank you for the comments. We added in the introduction the suggested information:

Page 2… Obesity is a frequent condition associated with NAFLD, and its prevalence is increasing, around 28.6% of the U.S. population (90 million obese in a population of 315 million). In the last decades, the incidence is increasing as well, higher in women (38.3%) than in men (34.3%). Worrying is the constant growth of obese children, that probably will represent the future population affected by NAFLD/NASH in more and more young people[3]. Models based on current incidence, it has been estimated that by 2030 almost 40% of the world's population will be overweight and 20% will be obese[4]. As a consequence, according to modern data and future projections, NAFLD is expected to increase to 33.5% among adults by 2030 (2), causing in the same time frame a higher rate of hepatocellular carcinoma (+137%) and liver-related deaths (+178%)[5]…

Page 2 and 3… The aim of the paper is to review and categorize the broad spectrum of available therapies for NASH/NAFLD, based on its pathogenesis. In addition, we will evaluate the role of bariatric surgery, and the impact of liver transplantation on the NAFLD outcome…

Page 3…The knowledge of the pharmacological targets comes from the understanding of the NAFLD pathogenesis. NAFLD genesis accounted for multiple factors. Liver inflammation and subsequent fibrosis are the key elements for NAFLD/NASH development. As for other diseases, “multiple hit” hypotheses for the development of NAFLD have been proposed[7]. An unhealthy lifestyle, leading to excessive caloric intake, insulin resistance, gut dysbiosis, visceral fat mass and increased hepatic de-novo lipogenesis and their interactions are all factors that have been related to NAFLD/NASH pathogenesis. Among them, insulin resistance seems to be an important element in the development of NASH. A genetic predisposition has been also proposed[8].

According to recent developments in molecular researches, we can summarize NAFLD’s pathogenesis in three major fields, involving adipose tissue, nutrients and intestine (See Figure 2). It is important to note that the knowledge of NAFLD’s pathogenesis is still under evaluation…

Can you add a discussion section where you can perhaps discuss the advantages and drawbacks of the treatment methods and provide a personal outlook of the future of treatment methods? 

Thank you for the suggestion. As proposed, we added a section with the conclusion where we discussed the current pro and cons of the available treatments.

Page 14 and 15

  1. Discussion and Conclusion

NAFLD is an increasing cause of morbidity and mortality, especially in the Western area. This disease is often associated with other important comorbidities, such as obesity, hypertension, DTM2, which makes its treatment particularly complex. To date, there is no gold standard pharmacological prevention and/or treatment for NAFLD, due to different reasons. First, the mechanisms underlying NAFLD are still only partially understood and pharmacological therapies are still not standardized. Second, many suggested drugs failed to improve the liver histological hallmarks of NAFLD. Third, a current problem is its early diagnosis. NAFLD is usually asymptomatic and, to date, we lack the tools that can accurately recognize the disease. As for other diseases, early detection is essential, in order to treat the condition before its complications and progression. Fourth, different therapeutic options are available, according to the stage of the disease. In particular, the treatment options can range from lifestyle modification, pharmacological options, and bariatric surgery ending to LT in selected cases. However, the general understanding about current treatments (pharmacological, surgical, or both) is that we lack evidence of significant benefit in terms of NAFLD improvement. It could be reasonable, from our perspective, to anticipate medical and surgical treatments usually started in later phases of the disease, like bariatric surgery in the setting of clinical trials.

These treatment options are not standardized, and the selection of the best ones and their interactions can be challenging. All of the available treatments should be individually tailored to the patient's needs and comorbidities, with a multi-disciplinary approach that considered lifestyle modification, the current pharmacological and surgical treatment options.

Round 2

Reviewer 1 Report

I would like to thank the authors for taking into account my comments.

Please rephrase : "Apoptosis is another important step towards NASH progression" by "Programmed cell death such as necroptosis is another step towards NASH progression". Indeed, one recent clinical trial (PMID 31887369) using apoptosis inhibitors has to shown to worsen liver injury in NASH patients by activating alternative cell death mode such as necroptosis.